# Delayed ART initiation in "Test and Treat era" and its associated factors among adults receiving antiretroviral therapy at public health institutions in Northwest Ethiopia: A multicenter cross-sectional study

Berihun Bantie[1]*, Gebrie Kassaw Yirga[1], Moges Wubneh Abate[1], Abreham Tsedalu Amare[1], Adane Birhanu Nigat[1], Agmasie Tigabu[1], Gashaw Kerebeh[2], Tigabu Desie Emiru[2], Nigusie Selomon Tibebu[2], Chalie Marew Tiruneh[2], Natnael Moges Misganaw[2], Dessie Temesgen[3], Molla Azmeraw Bizuayehu[3], Ahmed Nuru[4], Endalk Getasew Hiruy[5], Amare Kassaw[2]

1 Department of Comprehensive Nursing, College of Health Science, Debre Tabor University Northwest Ethiopia, Debra Tabor, Ethiopia, 2 Department of Pediatrics and Child Health, College of Health Science, Debre Tabor University, Debra Tabor, Northwest Ethiopia, 3 Department of Nursing, College of Health Science, Woldia University, Weldiya, Northeast Ethiopia, 4 Department of Nursing, College of Medicine and Health Science, Wolkite University, Welkite, Southern Ethiopia, 5 Department of Professional Nurse Specialty, Saint Peters Comprehensive Specialized Hospital, Addis Ababa, Ethiopia

* berihunbante@gmail.com

**Data Availability Statement:** All relevant data is contained within the paper and its Supporting information files. Hence, we uploaded a Supporting

## Abstract

### Background

Antiretroviral therapy (ART) has shown promising effects on the reduction of new HIV infection as well as HIV-related morbidity and mortality. In order to boost the effect of ART on ending HIV epidemics by 2030, the World Health Organization (WHO) indeed introduced a universal test and treat strategy in 2015 that recommends rapid (within seven days) initiation of ART for all HIV-positive patients. However, in low-income countries, a substantial number of HIV-positive patients were not enrolled in time, and information on delayed ART initiation status in Ethiopia is limited.

### Method

A multicenter cross-sectional study was conducted on 400 HIV-positive adults receiving ART at public health institutions in Bahir Dar city, Northwest Ethiopia. A structured checklist was used to extract data from the patient's medical record. Data was entered into Epi-data version 4.6 and exported to SPSS version 26 for further analysis. Both simple and multivariable binary logistic regressions were executed, and variables with a p-value < 0.05 in the final model were considered significant predictors of delayed ART initiation.

### Results

The magnitude of delayed ART initiation was 39% (95% CI: 34%–44%). Being male [Adjusted odds ratio(AOR) = 1.99, 95%CI:1.3–3.2], having opportunistic infections (OIs)

information file containing all the information used to affirm the findings described in the paper.

**Funding:** The author(s) received no specific funding for this work.

**Competing interests:** The authors have declared that no competing interests exist.

**Abbreviations:** AOR, Adjusted Odds Ratio; BMI, Body Mass Index; CD4, Cluster Differentiation cells; CI, confidence Interval; COR, Crude Odds Ratio; FMOH, Federal Ministry Of Health; IQR, Inter Quartile Range; OIs, Opportunistic Infections; PLHIV, Peoples Living with Human Immune Virus; ROC, receiver operating curve; SSA, Sub-Saharan African country; TB, Tuberculosis; UNAIDS, Joint United Nations Program on HIV/AIDS; UTT, Universal Test and Treat strategy; WHO, World Health Organization.

[AOR = 2.50, 95%CI:1.4–4.6], having other chronic diseases [AOR = 3.70,95%CI:1.7–8.3], substance abuse [AOR = 3.79, 95%CI: 1.9–7.4], having ambulatory functional status [AOR = 5.38, 95%CI: 1.4–9.6] and didn't have other HIV-positive family member [AOR = 1.85, 95%CI: 1.2–2.9] increases the odds of delayed ART initiation.

## Conclusion and recommendation

The burden of delayed ART initiation is found to be high. The presence of OIs and other chronic problems, substance abuse, ambulatory functional status, being male, and not having other HIV-positive family members were identified as significant predictors of delayed ART initiation. Special emphasis needs to be considered for those individuals with the identified risk factors.

## 1. Introduction

By the end of 2020, UNAIDS reported that nearly 37.7 million individuals were living with HIV/AIDS worldwide, of which 20.6 million (52.2%) reside in the Eastern and South African regions [1]. Ethiopia is one of the countries in this region that has a high HIV/AIDS-related morbidity and mortality rate, with over 670,000 people living with the disease and nearly 12,000 deaths by the end of 2019 [1, 2]. The combined use of ART has been the mainstay treatment and prevention option to control this catastrophic infection since the late 1980s. ART has also shown promising effects on the reduction of new HIV infections as well as HIV-related morbidity and mortality. According to the UNAIDS 2020 report, ART has contributed to a 30% and 42% reduction in new HIV infection and HIV-related mortality, respectively, since 2010 [1]. In order to boost the effect of ART on ending HIV epidemics by 2030, the World Health Organization (WHO) indeed launched a universal test and treat strategy in 2015 [3]. The universal test and treat (UTT) strategy is explained as the initiation of ART within seven days (including same-day) of confirmation of HIV status, regardless of the patient's WHO clinical stage or CD4+ count [3–5]. Furthermore, Ethiopia adopted this UTT strategy in 2016 and has begun to implement it since then [5]. The Federal Ministry of Health (FMOH) in Ethiopia immediately revised its comprehensive HIV care and treatment guidelines, and subsequent training was provided for the health care professionals to create a common understanding on the way forward, and it is now universally practiced throughout 1,224 ART-service providing health facilities in the country [5].

The rapid initiation of ART has been found to have numerous positive effects on reducing HIV-related morbidity and mortality. It plays a pivotal role in increasing access to ART, ensuring maximal and durable viral load suppression, restoring and preserving immune function, improving quality of life, and preventing further transmission of HIV/AIDS [6–10]. Despite such remarkable benefits, the magnitude of delayed ART initiation is high, particularly in low-income settings, which continues as a major challenge for the success of UNAIDS 2030 ending HIV epidemic goals [11]. A study conducted in Taiwan immediately after the endorsement of the UTT reported that nearly 68.3% of the newly diagnosed HIV-positive individuals were initiated into ART within seven days of confirmation of diagnosis [6]. In Africa, a study conducted in Ekurhuleni District, South Africa, revealed that only 54% of HIV-positive people were initiated into ART within seven days of confirmation of diagnosis [12]. Likewise, related studies in South Africa and Zimbabwe found that approximately 40.1% and 65% of HIV-positive patients were enrolled in ART on the same day as their diagnosis, indicating that there is still a problem

with early ART initiation [13, 14]. Though there is a paucity of evidence that directly investigates the burden of delayed initiation following the implementation of the UTT strategy in Ethiopia, a prior study conducted at Nekemte hospital, Western Ethiopia, showed that more than two-thirds (34%) of HIV-positive patients were enrolled in ART after seven days of diagnosis [15]. Another related study done in northwest Ethiopia revealed that only 41.9% of HIV-positive adults were initiated into ART on the "same day" of confirmation of diagnosis [16].

Delayed ART initiation is strongly associated with poor treatment outcomes, including lower immunologic response, poor virologic suppression, an increase in the incidence of OIs and other chronic problems, a high treatment failure rate, increased patient hospitalization, and, finally, an increase in the morbidity and mortality rate of HIV-positive patients [16–19]. People living with HIV/AIDS who didn't know other ART users, had tuberculosis and HIV (TB/HIV) co-infection, visited traditional healers, had CD4 count > 351 cells/mm3, and had normal body mass index (BMI) status were more likely to have enrolled into ART lately than their respective counterparts [15, 18]. Furthermore, residence, educational status, marital status, age, alcohol abuse, distance from ART center, perceived susceptibility and severity towards HIV/AIDS, and perceived home-based care status significantly affects the time to ART initiation [11, 15–18, 20]. To overcome this problem, solutions such as continuous professional development on HIV/AIDS management protocols, timely updating of management guidelines, scaling up ART centers, providing client-centered follow-up systems, formulating fixed-dose combination ART regimens, and providing continuous health education and promotion activities were proposed and implemented worldwide [4, 5, 21]. Indeed, the HIV continuum of care model, which includes activities such as HIV testing for identified risk groups, prompt linkage of HIV positive people to ART, and monitoring viral suppression rate, was utilized to track and evaluate the performance of the universal test and treat strategy [5].

Despite such efforts, a significant portion of HIV-positive patients in low and middle-income countries have not been timely engaged into ART, and data on delayed ART initiation status in Ethiopia is limited. Hence, this study is intended to fill the gap by estimating the magnitude of delayed ART initiation with its predictor variables.

## 2. Methods and materials

### 2.1 Study design and period

Institution-based cross-sectional study was conducted among 400 individuals who were enrolled into ART from November 2016 to December 2020 in public health institutions of Bahir Dar city. The city is the area of the earliest ART service center in Amhara regional state whereby it starts providing ART care since 2003 initially in Felegehiwot comprehensive specialized hospital. It is located at 565 km Northwest of Addis Ababa, the capital city of Ethiopia. Currently, the city has around 11 (two hospitals and nine health center) functional ART centers. These ART centers offer services including early HIV-testing of risk groups, providing ART for HIV-Positive individuals and continuous follow-up of patients' progress. According to the report of the Amhara Regional Health Bearou (ARHB) in Ethiopia, the city bears the highest burdens of HIV/AIDS patients load in the region by which nearly 14024 individuals were ever enrolled at those institutions for ART services since 2003. From the above figure, a total of 2822 HIV-positive patients were newly enrolled in to ART after the implementation of the treat-all strategy.

### 2.2. Study subjects

The source population for this study was all adults receiving ART after the test and treat era at public health institutions in Bahirdar city. The study population was all adults newly initiated

into ART from November 2016 to December 2020 at public health institutions in Bahirdar city. People living with HIV/AIDS (PLHIV) who had TB/HIV co-infection at presentation, Cryptococcus meningitis, and individuals with an incomplete record of HIV diagnosis and initiation date were excluded from the study.

## 2.3 Sample size determination and sampling technique

The minimum sample size required to conduct this study was calculated by double population proportion formula through EPINFO version 7.2.3.1 software. Parameters such as 80% power, 95% confidence level, the proportion of outcomes in the unexposed group (25.8%), the proportion of outcomes in the exposed group (39.5%), and an odds ratio of 1.87 [15] were used, and finally, a sample size of 400 was estimated to complete this study. Study participants were selected by computer generated simple random sampling technique. Initially, the unique ART number of eligible HIV-positive patients were gathered from all ART centers, then exported to version 10 Microsoft excel, and finally, 400 individuals were selected through computer generated simple random sampling technique.

## 2.4 Study variables

The main dependent variable for this study is the occurrence of delayed ART initiation. Whereas, variables like age, sex, level of education, marital status, occupation, residence, catchment area, have HIV positive family member, having cell phone status, baseline CD4 count, baseline WHO clinical stage, baseline BMI level, baseline functional status, presence of OIs, baseline substance abuse status, and HIV disclosure status of HIV-positive patients were categorized as independent predictor variables.

## 2.5 Operational definitions

Time to ART initiation is the time interval in a day from the confirmation of HIV status till the patient is enrolled into ART. Universal test and treat strategy is defined as the rapid(within seven days) initiation of ART irrespective of patients WHO clinical staging as well as $CD4^+$ cell counts [3, 5]. On the other hand, delayed ART initiation could be explained as the initiation of ART beyond seven days of confirmation of HIV status except for those who had either TB or Cryptococci meningitis opportunistic infections which requires differing of ART for two up to eight weeks to prevent serious adverse effects [4, 5]. The functional status of the patient is categorized as "working" if daily activities of Peoples living with HIV/AIDS was not altered due to illness, "ambulatory" if the patient was not fully working but was able to do minor tasks at home, and "bedridden" when the patient remained in bed most of the time [5]. BMI status was calculated by dividing body weight by height squared. It was classified as " Normal weight" when BMI level of patients is between (18.5–24.99 Kg/m$^2$), underweight (BMI <18.5 kg/m$^2$), and overweight (BMI ≥25kg/ m$^2$) [5]. Disclosure is recorded as "yes" when one's HIV positive status is informed to at least one individuals [5]. Individuals will be categorized as substance abused when the patient scores two or more on Cut down, Annoyed, Guilty, and Eye-opener (CAGE) substance abuse screening tool [22].

## 2.6 Data collection instrument and procedures

The data abstraction tool was adapted from the Ethiopian FMOH ART follow-up, the patient intake, and monitoring formats. The data was collected by three BSc Nurse Professionals under the close supervision of one master's in public health expert. To assure data quality, one day training was given for both data collectors and supervisors about the purpose of the study,

the way to extract relevant data, and ensure the confidentiality issue of the patient information. In addition, the consistency of the data abstraction checklist with the medical recording system was verified by using 10 randomly selected charts, and hence little modification was done on the formatting of the questionnaire.

## 2.7 Data processing and analysis procedures

The data was initially coded, entered into Epi data version 4.6.0 and simultaneously exported to SPSS version 26 software for further statistical analysis. Before proceeding to descriptive and inferential analysis of the data, missing records for variables like BMI, occupational status, educational status, and CD4 levels were handled and managed using the multiple imputations missing data management method. Then, a sensitivity analysis was executed to check the consistency of the output of the model with original and imputed data.

   The data was summarized using the median with interquartile range (IQR) for continuous variables and proportions for categorical variables, respectively. Moreover, tables and graphs were also used to present data. Both simple binary and multivariable binary logistic regression was fitted to identify the predictors of delayed ART initiation. A total of nine (09) variables with a P-value of < 0.25 in simple binary logistic regression were fitted to a multivariable binary logistic regression model, and finally, five variables with p-value of <0.05 were considered as significant predictors of delayed ART initiation. The backward likelihood ratio elimination method was used to select predictor variables by removing the least contributing variables until all variables left in the model are significantly associated with the outcome variable. "Hosmer and Lomeshow" goodness of fit test was used to assess the overall goodness of the model with the fitted data. In addition, the receiver operating characteristic curve (ROC curve), which is produced by plotting the sensitivity against 1-specificity (false positivity rate) of the model was used to check the discriminative power of the final model. The model was classified as "excellent" in discriminating patients with and without delayed ART initiation when the area under the curve (AUC) value in the ROC is ≥70%.

## 2.8 Ethical considerations

An ethical approval letter was obtained from the Institutional Review Board (IRB) of Bahirdar University, College of Medicine and Health Science with a protocol number of 058/2021. Because the study was conducted retrospectively from a patient's medical record, informed consent from patient was waived; instead, a supportive letter was sent from the IRB to each public health institution in Bahirar Dar City. Then a permission letter that ensures the use of patients' charts as a source of information was obtained from the directors and ART clinic focal person of those health institutions, respectively. During data collection and entry, patient identifiers (patient's medical registration number (MRN)) were replaced by new identification numbers or codes. Besides, the collected data was treated with confidentiality.

## 3. Result

### 3.1 Socio demographic characteristics

A total of 400 HIV-positive adults who were enrolled in ART following Test and Treat strategy were included for final analysis. Of these, more than half of them were females (57.3%), married (51.5%), and had HIV-positive family members (59.0%). The median age of the study participants was 32 years, with an interquartile range (IQR 40–27) of years. Regarding to their residence, around 229 (57.3%) of individuals reside beyond their catchment area (Table 1).

**Table 1. Socio-demographic characteristics adults receiving ART at public health institutions in Bahir Dar city, Northwest Ethiopia,2022.**

| Variables (N = 400) | Category | Total frequency | % |
|---|---|---|---|
| Sex | Male | 171 | 42.8 |
| | Female | 229 | 57.3 |
| Age category | Age 15–24 | 61 | 15.2 |
| | Age 25–34 | 176 | 44.0 |
| | Age 35–45 | 128 | 32.0 |
| | Age >45 | 35 | 8.8 |
| Marital status | Married | 206 | 51.5 |
| | Not Married | 194 | 48.5 |
| Educational status | No formal education | 117 | 29.3 |
| | Primary education | 99 | 24.7 |
| | Secondary | 106 | 26.5 |
| | Tertiary and above | 78 | 19.5 |
| Occupation | Daily-laborer | 82 | 20.5 |
| | Farmer | 24 | 6.0 |
| | Merchant | 59 | 14.8 |
| | House-wife | 67 | 16.8 |
| | Employed | 119 | 29.8 |
| | Student | 23 | 5.8 |
| | [1]Others | 26 | 6.5 |
| Had positive family member | Yes | 164 | 41.0 |
| | No | 236 | 59.0 |
| Reside on catchment area | Yes | 229 | 57.3 |
| | No | 171 | 42.7 |
| Having cell-phone | Yes | 358 | 89.5 |
| | No | 42 | 10.5 |

Others:- include those individuals who are drivers and do not have jobs, N- sample size and %- Indicates column percentage

## 3.2 Clinical, treatment-related and behavioral characteristics

The median CD4 and BMI status of the study participants during their enrollment to ART was 316 cell/mm3 with an interquartile range (IQR 502- 148cell/mm3) and 20.42 with IQR (22.67–18.59) respectively. On the other hand, nearly 13% of individuals abused at least one of the common substances, including chat, cigarettes, and alcohol. About one fourth (26%) of the study participants had at least one OIs. Indeed, a substantial number of individuals (8.5%) had at least one additional chronic coexisting problem (Table 2).

## 3.3 Magnitude of delayed ART initiation

A total of 156 (39%) HIV-positive adults were categorized as having delayed ART initiation (Fig 1). However, the proportion of delayed ART initiation varied substantially across HIV-positive adults based on their socio-demographic, clinical, and treatment-related characteristics.

## 3.4 Predictors of delayed ART initiation

In the final multivariable logistic regression model, at 5% of the level of significance, the following six variables, namely sex, having HIV-positive family member, substance abuse,

**Table 2. Clinical and behavioral characteristics of adults receiving ART at public health institutions in Bahir Dar city, Northwest Ethiopia, 2022.**

| Variables (N = 400) | Categories of a variable | Total frequency | % |
|---|---|---|---|
| Baseline BMI Status | Under weight | 94 | 23.5 |
| | Normal weight | 270 | 67.5 |
| | Over weight | 36 | 9.0 |
| Baseline CD4+ count | < 200 | 118 | 29.5 |
| | From 200–499 | 149 | 37.3 |
| | Greater than 500 | 133 | 33.3 |
| Baseline WHO clinical stage | WHO stage I&II | 308 | 77.0 |
| | WHO stage III&IV | 92 | 23.0 |
| functional status | Working | 317 | 79.3 |
| | Ambulatory | 66 | 16.5 |
| | Bedridden | 17 | 4.3 |
| History of OIs | Yes | 104 | 26.0 |
| | No | 296 | 74.0 |
| Presence of Chronic diseases | Yes | 34 | 8.5 |
| | NO | 366 | 91.5 |
| Substance abuse status | Yes | 54 | 13.5 |
| | No | 346 | 86.5 |

%- indicates column percentage, BMI- Body Mass Index, OIs- opportunistic infections

chronic disease, OIs, and functional status of the patients, were found to be significant predictors of delayed ART initiation. Accordingly, the odds of delayed ART initiation is nearly two times in males than females (AOR 1.78:- 95% CI 1.13–2.8). HIV positive individuals who didn't have at least one HIV positive family member had nearly two or more odds of delayed

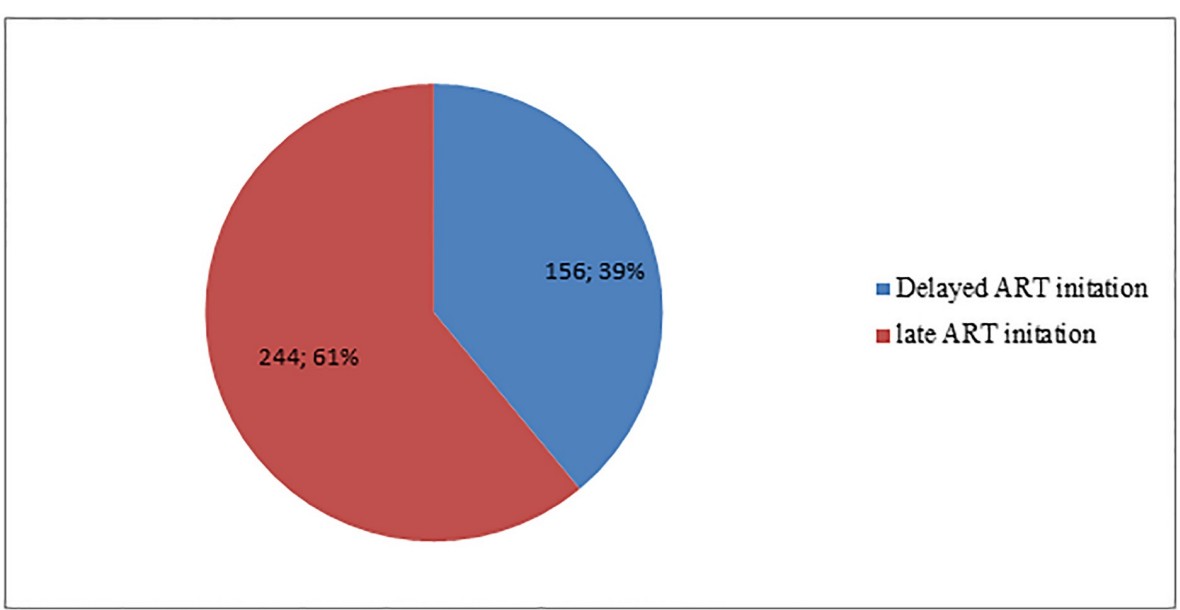

**Fig 1. Delayed ART initiation status of adults receiving ART following Test and Treat strategy at public health institutions in Bahir Dar city, Northwest Ethiopia, 2022.**

ART initiation than individuals who knew at least one family member. The odds of delayed ART initiation among individuals who had at least one other chronic diseases was 3.7 times (AOR 3.7, 95% CI:1.7–8.3) than individuals with no other chronic diseases. Substance abused HIV-positive individuals were 3.79 times more at risk of having delayed ART initiation than their counterparts (AOR 3.79, 95%CI: 1.9–7.4). The odds of delayed ART initiation among individuals with ambulatory functional status (AOR 5.38, 95%CI: 1.4–9.6) was nearly five-times greater than among individuals with working functional status (Table 3).

### 3.5 Assumption of binary logistic regression

Hosmer and Lomeshow goodness of test for multi logistic regression revealed that the model constituting such variables is good to predict the outcome variable i.e delayed ART initiation (P-value = 0.639). Indeed the discrimination power of the model was checked by ROC analysis and the area under the curve value **(AUC value = 0.744 (74%))** illustrated that the model is excellent to discriminate individuals with delayed ART initiation (true positives) and individuals without delayed ART initiation (Fig 2).

## 4. Discussion

There is a dearth of studies that address the time of ART initiation following the implementation of the Test and Treat strategy in Ethiopia. This study has found that about 39% (95%CI: 34–44%) of HIV-positive adults were initiated into ART beyond seven days. This finding is in line with recent studies conducted in Nekemete, Western Ethiopia [15], Ekurhuleni district in South Africa [12], On the other hand, this finding is slightly lower than a study conducted in Taiwan [6], which might be due to a difference in socioeconomic and educational status between the two countries.

In the current study, male HIV-positive individuals had higher odds of being delayed for ART initiation than their counterparts. In the current study, male HIV-positive individuals had a higher odds of being delayed for ART initiation than their counterparts. This finding is congruent with previous studies conducted in Nekemte [15] and South Africa [13]. The possible elucidation for this might be the fact that women's decision to immediately seek ART will be highly influenced by physician's counseling than males [23]. On top of the above, there is a claim that giving special attention to maternal and child health services by both global and national health policies may result in females getting stronger counseling, education, and monitoring by health care providers when compared to males, and this finally might influence women's decision to early engage in ART.

When compared to patients without documented baseline OIs, HIV-positive people with OIs had a two-fold increased odds of delaying ART initiation time. This finding is in parallel with prior studies conducted at South Wollo Zone and Nekemte Hospital in Ethiopia [15, 24]. Because of worries about potential drug interactions, toxicities, poor adherence owing to pill burden, and systemic inflammatory reconstitution syndrome (SIRS), both the patient and the healthcare provider may decide to delay ART initiation. Nevertheless, the latest evidence and guidelines recommend the immediate initiation of ART except for patients with TB/HIV co-infection and Cryptococcus meningitis [5].

The risk of delayed ART commencement is nearly four times greater among HIV-positive individuals with chronic problems than their counterparts. This finding is in line with previous systematic review and meta-analysis studies conducted before the current test -and -treat era in Ethiopia [20]. The possible justification for this might also be due to unjustified fears by patients and even by health professionals about drug-drug interactions, high pill burdens, and toxicities happening as a result of treating both chronic diseases simultaneously. This implies

**Table 3. Bivariable and multivariable logistic regression analysis result of delayed ART initiation among adult receiving ART at public health institutions in Bahir Dar city, Northwest Ethiopia, 2022.**

| Variable (N = 400) | Category | Outcome status | | COR [95% CI] | AOR [95%CI] | p-value |
|---|---|---|---|---|---|---|
| | | Delayed initiation | Early initiated | | | |
| Sex | Male | 83 | 88 | 2.02 (1.34–3.03) * | **1.99(1.26–3.16)**** | 0.01 |
| | Female | 73 | 156 | 1 | 1 | |
| Age category | Age 15–24 | 21 | 38 | 0.39 (0.25–1.71) | | |
| | Age 25–34 | 63 | 115 | 0.65 (0.27–1.53) | | |
| | Age 35–45 | 61 | 78 | 0.92 (0.38–2.2) | | |
| | Age >45 | 11 | 13 | 1 | | |
| Marital status | Married | 78 | 128 | 1 | | |
| | Not Married | 78 | 116 | 0.63 (0.74–1.64) | | |
| Educational status | No formal education | 47 | 70 | 0.81(0.56–1.93) | | |
| | Primary education | 42 | 57 | 0.59 (0.64–2.16) | | |
| | Secondary | 37 | 69 | 0.62 (0.47–1.57) | | |
| | Tertiary and above | 30 | 48 | 1 | | |
| Occupation | Daily-laborer | 24 | 58 | 1 | | |
| | Farmer | 14 | 10 | 3.38 (1.32–8.67)* | | |
| | Merchant | 26 | 33 | 1.9 (0.95–3.84) | | |
| | House-wife | 26 | 41 | 1.53 (0.77–3.03) | | |
| | Employed | 44 | 75 | 1.42(0.78 2.59) | | |
| | Student | 11 | 12 | 2.21(0.86–5.70) | | |
| | Others | 11 | 15 | 1.77(0.71–4.41) | | |
| Had HIV-positive family member | Yes | 48 | 116 | 1 | 1 | |
| | No | 108 | 128 | 2.04 (1.33–3.11)* | **1.85(1.2–2.9)**** | 0.01 |
| Reside on catchment area | Yes | 148 | 81 | 1 | | |
| | No | 96 | 75 | 1.42 (0.47–1.05) | | |
| Having Cell _phone | Yes | 214 | 144 | 1 | 1 | |
| | No | 30 | 12 | 0.59 (0.29–1.12) | 0.5 (0.2–1.11) | 0.09 |
| Baseline BMI status | Under weight | 57 | 37 | 1 | | |
| | normal weight | 161 | 109 | 1.04 (.645–1.685 | | |
| | Overweight | 26 | 10 | 0.59 (0.26–1.370) | | |
| Baseline CD4+ count | < 200 | 43 | 75 | 1 | | |
| | From 200–499 | 54 | 95 | 0.99 (0.6–1.64) | | |
| | Greater than 500 | 59 | 74 | 1.39 (0.837–2.31) | | |
| Baseline WHO clinical stage | Stage I and II | 115 | 195 | 0.67 (0.41–1.06) | | |
| | Stage III and IV | 41 | 49 | 1 | | |
| Functional status | Working | 136 | 198 | 1 | 1 | |
| | Ambulatory | 14 | 41 | 1.82 (0.63–5.3) | **5.38(1.4–9.6)**** | 0.01 |
| | Bedridden | 6 | 5 | 1.82 (0.46–7.12) | 0.95 (0.2–3.67) | 0.94 |
| History of OIs | NO | 106 | 190 | 1 | 1 | |
| | Yes | 50 | 54 | 1.66 (1.05–2.6)** | **2.5 (1.4–4.6)**** | 0.03 |
| Presence of Chronic diseases | NO | 133 | 233 | 1 | 1 | |
| | **Yes** | 23 | 11 | 3.67(1.73–7.75) * | **3.7(1.7–8.3)** ** | 0.01 |
| Substance Abuse | Yes | 225 | 125 | 1 | 1 | |
| | No | 22 | 35 | 3.42 (1.88–6.24) * | **3.79(1.9–7.4)**** | 0.01 |

1-Reference category, COR -crude odds ratio, AOR-adjusted odds ratio, CI- confidence interval,

*—statistically significant with simple binary logistic regression,

** -statistically significant at multivariable regression model

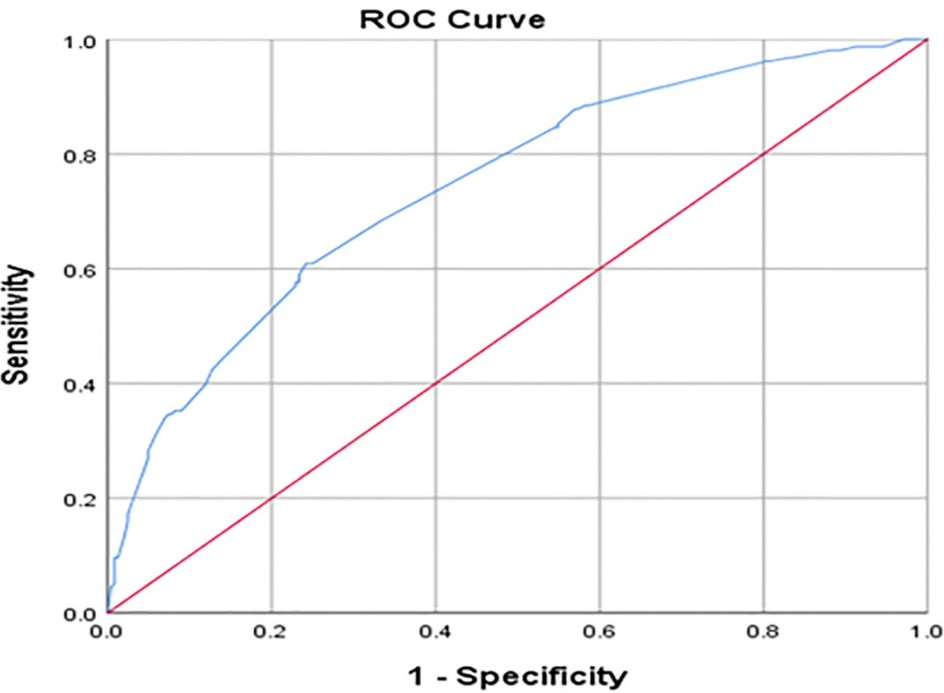

**Fig 2. ROC curve graph showing model discrimination power towards delayed ART initiation among adults receiving ART at public health institutions in Bahir Dar city, Northwest Ethiopia, 2022.**

that healthcare providers' experience in the management of chronic diseases and comorbidities of HIV/AIDS should be continuously updated, so patients may receive adequate education and counseling about it. This further implies that continuous professional development of healthcare providers through expanding on-the-job training, and short-term and long-term capacity-building sessions needs to be strengthened.

The odds of delayed ART initiation among individuals who don't have other HIV-positive family member are higher than individuals with other HIV-positive family member. The possible elucidation of this link might be due to the fact that HIV-positive people with other HIV-positive family members may get better reinforcement and social support and then build confidence to engage in ART sooner than their counterparts [25].

HIV-positive individuals who were abused for at least one of the common substances (chat, alcohol, and cigarettes) had higher odds of being late on ART initiation. The possible explanation for this finding might be due to the fact that substance abuse has an overwhelming effect on each aspect of HIV/AIDS. Substance abusers, for example, will experience cognitive impairment, which will further hamper their decision-making abilities and rigorous adherence to their ART therapy. The finding is consistent with literature conducted in the previous era in South Wollo and Myanmar [24, 26].

The time it takes for HIV-positive people to start taking antiretroviral therapy (ART) has a significant association with their baseline functional status. This is congruent with a study done in South Africa [13]. Patients who were ambulatory, for example, had a higher chance of being delayed in receiving ART when compared with working individuals. It could be driven by the fact that HIV-positive individuals with working functional status believe they are

healthy enough and that initiating antiretroviral therapy (ART) while they are relatively healthy may not be beneficial.

## 4.1 Limitation of the study

We acknowledge that our findings should be considered in light of the following limitations: Because of the incompleteness of the records, we can't measure the effect of healthcare provider and institution-level factors on the ART initiation time of HIV-positive adults. The data was collected at a single point in time; a lack of temporal relationship might happen between some predictor variables and the outcome variable.

## 5. Conclusion and recommendation

In this test-and-treat era, the magnitude of delayed ART initiation is significantly high. Males, in particular, had a high rate of delayed ART initiation compared to their counterparts. Moreover, the presence of OIs and chronic problems, substance abuse, ambulatory functional status and not knowing other HIV-positive family members were identified as significant predictors of late ART initiation. This study highlighted the importance of providing intensive counseling and education about the early invitation of ART for those risk groups. Further research shall be undertaken to explore the effect of other socio-economical, behavioral, institutional, and contextual factors on delayed ART initiation.

## Supporting information

**S1 Data.**
(RAR)

## Acknowledgments

We would like to express our deepest gratitude to Bahirdar University, College of Medicine and Health Science, for letting us conduct this research and for timely approving ethical clearance. Next, we also extend our profound thanks to the staff working at the ART clinic, card room, and administration positions in selected public health institutions in Bahirdar town, facilitating a conducive environment to obtain the required data for this research. Last but not least, our special thanks go to the data collectors and the supervisor for their willingness, commitment, and intensive work during data collection for the betterment of this research.

## Author Contributions

**Conceptualization:** Berihun Bantie.

**Data curation:** Berihun Bantie, Gebrie Kassaw Yirga, Moges Wubneh Abate, Nigusie Selomon Tibebu, Chalie Marew Tiruneh, Natnael Moges Misganaw, Ahmed Nuru, Endalk Getasew Hiruy, Amare Kassaw.

**Formal analysis:** Berihun Bantie, Gebrie Kassaw Yirga, Moges Wubneh Abate, Abreham Tsedalu Amare, Adane Birhanu Nigat, Agmasie Tigabu, Gashaw Kerebeh, Tigabu Desie Emiru, Nigusie Selomon Tibebu, Chalie Marew Tiruneh, Natnael Moges Misganaw, Dessie Temesgen, Molla Azmeraw Bizuayehu, Ahmed Nuru, Endalk Getasew Hiruy.

**Funding acquisition:** Endalk Getasew Hiruy.

**Investigation:** Berihun Bantie, Gebrie Kassaw Yirga, Moges Wubneh Abate, Abreham Tsedalu Amare, Adane Birhanu Nigat, Agmasie Tigabu, Gashaw Kerebeh, Tigabu Desie Emiru,

Nigusie Selomon Tibebu, Chalie Marew Tiruneh, Natnael Moges Misganaw, Dessie Temesgen, Molla Azmeraw Bizuayehu, Ahmed Nuru, Amare Kassaw.

**Methodology:** Berihun Bantie, Gebrie Kassaw Yirga, Moges Wubneh Abate, Abreham Tsedalu Amare, Adane Birhanu Nigat, Agmasie Tigabu, Gashaw Kerebeh, Tigabu Desie Emiru, Nigusie Selomon Tibebu, Natnael Moges Misganaw, Dessie Temesgen, Molla Azmeraw Bizuayehu, Ahmed Nuru, Endalk Getasew Hiruy, Amare Kassaw.

**Resources:** Berihun Bantie.

**Software:** Berihun Bantie, Gebrie Kassaw Yirga, Adane Birhanu Nigat, Agmasie Tigabu, Gashaw Kerebeh, Tigabu Desie Emiru, Nigusie Selomon Tibebu, Chalie Marew Tiruneh, Natnael Moges Misganaw, Dessie Temesgen, Molla Azmeraw Bizuayehu, Ahmed Nuru, Endalk Getasew Hiruy, Amare Kassaw.

**Supervision:** Berihun Bantie, Adane Birhanu Nigat, Chalie Marew Tiruneh, Molla Azmeraw Bizuayehu.

**Validation:** Moges Wubneh Abate, Abreham Tsedalu Amare, Adane Birhanu Nigat, Nigusie Selomon Tibebu, Ahmed Nuru, Endalk Getasew Hiruy.

**Writing – original draft:** Berihun Bantie, Gebrie Kassaw Yirga, Moges Wubneh Abate, Abreham Tsedalu Amare, Adane Birhanu Nigat, Agmasie Tigabu, Gashaw Kerebeh, Tigabu Desie Emiru, Nigusie Selomon Tibebu, Chalie Marew Tiruneh, Natnael Moges Misganaw, Dessie Temesgen, Molla Azmeraw Bizuayehu, Ahmed Nuru, Amare Kassaw.

**Writing – review & editing:** Berihun Bantie, Gebrie Kassaw Yirga, Moges Wubneh Abate, Abreham Tsedalu Amare, Adane Birhanu Nigat, Agmasie Tigabu, Gashaw Kerebeh, Tigabu Desie Emiru, Nigusie Selomon Tibebu, Chalie Marew Tiruneh, Natnael Moges Misganaw, Dessie Temesgen, Molla Azmeraw Bizuayehu, Ahmed Nuru.

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
