## [Decision Letter · Decision Letter 0]

7 Mar 2022

PONE-D-21-39787Delayed ART initiation in “Test and Treat era” and its associated factors among adults receiving ART at Public health institution in Northwest Ethiopia: A multicenter cross-sectional studyPLOS ONE

Dear Dr. Tesema,

Thank you for submitting your manuscript to PLOS ONE. After careful consideration, we feel that it has merit but does not fully meet PLOS ONE’s publication criteria as it currently stands. Therefore, we invite you to submit a revised version of the manuscript that addresses the points raised during the review process.

We look forward to receiving your revised manuscript.

Kind regards,

Myo Minn Oo, M.D., Ph.D.

Academic Editor

PLOS ONE

Journal Requirements:

Additional Editor Comments:

This investigation has some value and the findings are worthy of broader dissemination. However the English is very poor and this severely compromises our ability to fully understand and assess your paper. This is not simply a question of grammar and punctuation. There are many methodological inconsistencies and ambiguities to which Reviewer #2, in particular, has drawn your attention. The reviewers' comments are appended below.

We understand that English is not the first language of the authors however the paper requires a major overhaul and edit by someone with excellent English writing skills who is also familiar with the topic, the scientific methods and the publication requirements. We ask that you adhere to our Authors Guidelines.

Taking all these comments into account, we would like to invite you to revise your manuscript. However this invitation does not imply that your paper will be accepted.

Reviewers' comments:

Reviewer's Responses to Questions

**Comments to the Author**

1. Is the manuscript technically sound, and do the data support the conclusions?

Reviewer #1: Yes

Reviewer #2: Yes

Reviewer #3: Partly

2. Has the statistical analysis been performed appropriately and rigorously? 

Reviewer #1: Yes

Reviewer #2: Yes

Reviewer #3: Yes

3. Have the authors made all data underlying the findings in their manuscript fully available?

Reviewer #1: No

Reviewer #2: Yes

Reviewer #3: No

4. Is the manuscript presented in an intelligible fashion and written in standard English?

Reviewer #1: Yes

Reviewer #2: No

Reviewer #3: Yes

5. Review Comments to the Author

Reviewer #1: Abstract:

1) Please avoid abbreviations in abstract (e.g. "G.C" in line number 3)

2) inconsistent writing pattern of 95% CI (some have hyphen and some doesn't have hyphen between the two values)

3) Better to mention what is delayed ART initiation in abstract (e.g. delayed (ART initiation more than seven days from HIV confirmation))

Introduction:

1) Need a brief backgroud or context of the study sites, such as when was the policy for test and treat endorsed in ethiopia, current practice and procedures for test & treat strategy and challenges... also about a brief on HIV epidemic of ethiopia as well as study sites.

2) line number 33, please use correct tense for "define"

3) line number 41, what is meant by the phrase "ART rapidly" ? same day or within 7 days?

Methods and Materials

1) line number 116, please write more about data verification: need to write how to sample these 10 charts (e.g. random picking or something else?)

2) Data processing and analysis procedure : Need to tell if there was missing data or not and how missing data were handled ?

Results

1) line number 142: what is meant by "positive" , please re-phrase as "HIV positive" to be more specific

2) Table 1 and Table 2: To add foot note about % whether it is a row percentage or column percentage

3) line number 151: writing pattern of IQR should be consistent with others (e.g. Q1 - Q3). in line 151 the pattern is Q3 - Q1

4) line number 154: use of wording "significant number" should be rephrased. We can't say significant by seeing only the absolute numbers and percentages only

5) line number 180 & 181 : the reference is working functional status , so, it is better to rephrase this sentence (it is written bedridden as reference)

Figure 2 Title:

To write long form of FHCSH as figures should be stand alone and abbreviation should not be used.

Both Figures and Tables

- to add footnotes for every abbreviations used in figures and tables

Reviewer #2: Manuscript Number: PONE-D-21-39787

Manuscript Title: Delayed ART initiation in “Test and Treat era” and its associated factors among adults receiving ART at Public health institution in Northwest Ethiopia: A multicenter cross-sectional study

Abstract

Page 2, line 2: The background explanation is different from the first page of article.

Page 2, line7: In the patient’s chart, is HIV testing date is available? It is not clearly mentioned.

Page 2, line 14: Definition of chronic problem is not clear. Why social support is recommended?

Introduction

It would be better to include HIV morbidity and mortality data of Ethiopia.

Page 3, line 43 & Page 4, line 47: Reference 10,11 & 13, majority of patients received ART initiation within a day of diagnosis confirmed. May need to revise the sentences to show still need to reach the global target because in this study delay is defined for >7 days of diagnosis

Methods and Materials

It would be better to explain ART diagnosis and ART initiation processes. It can be performed at the same center or is there any patient take ART test from different places enrolled into ART service center? Can delay in ART initiation happen due to service factors?

Study Design and Period

Page 4, line 67: The author mentioned that study participants were 400 participants enrolled into ART from November 2016 to October 2020. But in page, line 78, the study population was selected from the record from November 2016 to December 2017. The study inclusion period is different.

Study Subjects

Page5, line 79: Why study subjects were selected from two years periods 2016-2017? Is there any changes in ART health care system after 2017? Using 2016-2017, the result may not reflect to the current situations? Adult is >15 years?

Page 5, line 80: Why TB/HIV coinfection at presentation, Cryptococcus meningitis infection are excluded from the study?

Sample size determination and Sampling technique

Page 5, line 85: It would be better to include the reference of using 25.8% proportion of outcome in unexposed group and Odds ratio of 1.87 for sample size calculation.

Operational Definitions

Page 6, line 98: It would be better to use delay initiation of ART for outcome definition.

 

Result

The name of outcome variable “Delayed ART initiation” like title should be better to use constantly in tables and texts.

Line 160, page 10: In Table 1 and Table 2, only overall frequency and percentage of all variables should be better to be mentioned without showing the classification of delay and early initiation of ART. Because data is duplicated with Table 3. (OR) Otherwise, the information of Prevalence of late ART initiation should be better presented first in the result before the Sociodemographic characteristics of participants. What is median time to ART initiation?

In Table 1, Had positive family member or know other HIV positive family member??

In Table 2, categorization of BMI is underweight or undernourished?? It would be better to use the same. Presence of Chronic Problems or chronic health problems?? It is not clear. Need foot note for BMI, OI etc

In Table 2 and table 3, Presence of chronic problem and History of chronic diseases?? Variables names are changed. It would be better to use constant variable name through the paper.

In Table 1 and table 3, Had positive family member or HIV-positive family member or know other HIV positive family member?? It would be better to use constant variable name through the paper.

In Table 1 and table 3, Reside on Catchment area or Catchment area Residence?? It would be better to use constant variable name through the paper.

Serial of variable names in Table 3 should be the same with that in Table 1 and Table 2. In table 3, Marital status should be before education like in Table 1. The same for Reside on Catchment area. In table 3 bivariate association of BMI and delay was not included.

In Table 3, what is ** in COR and what is COR and AOR? “-“signs should be shown between numbers showing 95% CI.

Table 3, In variables of HIV-positive family member, History of OIs, History of chronic diseases and Substance, first row should be used for reference by changing row positions. It is difficult to interpret the data. It can lead to misinterpretation.

Figure 2. How ROC curved is developed and how it is useful for your study should be written in data analysis portion. It should not be 2021 in Figure 2 title. Page 12 line 189, AOC or AUC??

Page 10, line 160: Prevalence of late ART initiation: It is better to use constant outcome variable name ”Delayed ART initiation”. Prevalence?

In Figure 1, I think I should not be 2021. Name of outcome variable should be constant with that in title. If median time taken and proportion of delay in ART initiation are shown in texts, figure 1 may not need.

Page 10, line 168: What is Predictors of loss to follow up?

Discussion

May to need to discuss about generalizability of the result. Why social support is recommended for reducing delay in this study? The result can reflect to the current situation?

Reviewer #3: conclusion is partly okay, but need much more evidence for what the authors brought up in the discussion session : some reasons / sentences went quite off. See line 211,226, 228, 240 (Please see the attached reviewed file for more details). Discussion and conclusion would be better if the authors can point out the limitations and generalisability of their research. Though the statistical analyses were performed appropriately and rigorously, the writing needs to be modified - for eg., instead of the word "bivariable logistic regression", it will be much correct to use "simple binary logistic regression for each potential predictor". Should have provided more details in the statistical analyses section so that the readers could understand the backward elimination process before going deep dive into the result section.

In this draft, the authors claimed that all relevant data are within the manuscript - which is true, but the data aren't fully available - this totally makes sense as this is a study of PLHIV where the study participants' confidentiality and privacy must be valued / taken care of.

There are some grammar mistakes and spelling errors, and found some necessary words and/or symbols dropped out of the sentences. So, the English of this draft will be acceptable after fixing up those mistakes/errors. (Please see the attached file for more details).

6. PLOS authors have the option to publish the peer review history of their article (what does this mean?). If published, this will include your full peer review and any attached files.

Reviewer #1: No

Reviewer #2: **Yes: **Kyaw Ko Ko Htet

Reviewer #3: **Yes: **Nilar Aye Tun

---

## [Author Response · Author response to Decision Letter 0]

6 Apr 2022

A rebuttal letter which address the points raised by academic editor and reviewers 

Manuscript Number: PONE-D-21-39787

Manuscript Title: Delayed ART initiation in “Test and Treat era” and its associated factors among adults receiving ART at Public health institution in Northwest Ethiopia: A multicenter cross-sectional study

1.Comments and their responses for points raised by academic editor

1. Please ensure that your manuscript meets PLOS-ONE’s style requirements, including for file names?

We the authors thoroughly observe PLOS –ONE formatting .pdf files and try to comply with it through the whole manuscript 

2. Please provide additional details regarding participants consent. in the ethics statement in methods and online submission forms ,please ensure you have specified what type of consent you obtained .

As noted in the method section of the abstract, the research is conducted ART service providing institution by using patient’s medical record (secondary data) as a source of information for data collection. Since we didn.t collect the information directly from patients, consent directly from patients is waived. Whereas, Ethical approval letter is obtained from IRB of Bahiradr university, followed by a permission letter which ensures the use of patients chart as source of information was obtained from director and ART focal-person of each sites. Then the data is collected from patients chart using the permission letter. ( page 9, line# 176)

3. In your Data Availability statement, you have not specified where the minimal data set underlying the results described in your manuscript can be found?

Answer- Since the study is conducted on peoples living with HIV AIDS (PLHIV), we share the minimal data set by keeping potentially identifier of the patient to keep confidential (Minimal data set file.

4. Please Amend either the abstract on the online submission form or the abstract in the manuscript. 

Response- We accept and made them uniform 

 2. Reviewer #1 . Abstract 

1. Please avoid abbreviation like G.C in Abstract -- Corrected as suggested ,Page 2, line#4-5

2. inconsistent writing pattern of 95% CI (some have hyphen and some doesn't have hyphen between the two values)- Corrected as suggested , Page 2,line# 16-19

3. Better to mention what is “delayed ART intimation in the abstract)- In order to maintain the flow of idea with the previous sentence , we prefer to explain about the “ test and treat strategy” including its definition i.e individuals will easily comprehend the initiation of the study. --- page 2#, Line # 3 -6 

Introduction section

1. Need a brief background or context of the study sites, such as when was the policy for test and treat endorsed in Ethiopia, current practice and procedures for test & treat strategy and challenges. Also about a brief on HIV epidemic of Ethiopia as well as study sites.? Answer- Accepted and we try to incorporate such important ideas on introduction part , Page 3, Line# 30-32, 41-46,

2. In line # 33 ,please use the correct word for “define”- 

Answer- Accepted and modified accordingly , page3, Line# 39

3. ) line number 41, what is meant by the phrase "ART rapidly" ? same day or within 7 days?

Response- Accepted and modified as “within seven days of confirmation of diagnosis , page 4, line # 59

Method and material section

1. Line number 116, please write more about data verification: need to write how to sample these 10 charts (e.g. random picking or something else?)---

Response:- In order to check the consistency between the data abstraction checklist and recording system on patients chart , we use 10 randomly selected charts which are excluded for the final study , page 8,line#151

2. Need to tell if there was missing data or not and how missing data were handled ?)- Response- Accepted and we just incorporate how we handle missed data. Before proceeding to descriptive and inferential analysis of the data, missed records for variables like BMI, occupational status educational status, and CD4 levels were handled and managed. Initially, we observe the missing patterns of the variables and we conclude that there is no clear missing pattern, hence we use multiple imputations missing data management method and impute those data’s accordingly. Then, sensitivity analysis was executed to check the consistency of the output of the model with original and imputed data ( Page 8, 155-159)

 Result section

1. line number 142: what is meant by "positive" , please re-phrase as "HIV positive" to be more specific 

Response- rephrased as HIV-positive, page 10, line 189

2. Table 1 and Table 2: To add foot note about % whether it is a row percentage or column percentage- 

Response - Accepted and modified accordingly including %( which it indicates the column %), table 1 on page 10, table 2 on page 11

3. Line#151, writing pattern of IQR is not consistent- 

Response- Accepted and corrected accordingly, page 10 line 190, page 11 line 197 

4. Line number 154: use of wording "significant number" should be rephrased. We can't say significant by seeing only the absolute numbers and percentages only- 

Answer- Accepted and rephrased it as “substantial number of individuals” ( page 11, line 200)

5. line number 180 & 181 : the reference is working functional status , so, it is better to rephrase this sentence (it is written bedridden as reference)- 

Answer-Accepted and reference for both of them is working functional status ( Table3, page 14)

 Figure 

To write long form of FHCSH as figures should be stand alone and abbreviation should not be used.- Response- Accepted and modified accordingly , Figure 02, page 15

Figures and footnote- 

to add footnotes for every abbreviations used in figures and tables - 

Response- Accepted and foot notes are added in each figure and table , (Table 1.2.3 on page 10,11 and 13 

3. Reviewer # 2. 

 Abstract section

1. Page 2, line 2: The background explanation is different from the first page of article) – 

Response- we accept it, and we simultaneously try to make the two ideas consistent. Currently, the background of the abstract almost compatible with the first paragraph of the introduction. Both of them tell about the effects of ART on HIV/ AIDS and its initiation time in this test and treat era. – (Page 2 line 3 and page 3, line 39-43)

2. Page 2, line 7( Page 2, line7: In the patient’s chart, is HIV testing date is available? It is not clearly mentioned.) - 

Response - Yes, the patient medical record( chart) includes history sheet, progress sheet, nursing sheet and ART intake and monitoring formats where by Patients HIV testing data and result are mainly documented in history sheet as well as ART intake and motoring form ( page 2,line 13) . 

3. Page 2, line 14 (Page 2, line 14: Definition of chronic problem is not clear. Why social support is recommended?)- 

Answer- Chronic problem is modified in to chronic diseases. Patient is said to has chronic diseases whenever they have a documented history one or more of the following diseases such as, diabetes mellitus, Hypertension (HTN), chronic heart dieses (CHD) , Stroke Chronic obstructive pulmonary diseases(COPD),asthma, chronic liver and kidney diseases and cancer in any of the body parts.

Why social support is recommended- 

Response - The recommendation was modified as ‘ Special emphasis shall be considered for individuals with identified risk factors in order to make recommendation short and concise. However, we initially recommend social support because as it is indicated on discussion part, the likely explanation for delayed initiation for those individuals who didn’t have other HIV-positive family member and who is abuse of substance is lack of social support from the family as well as the significant others . ( page 3,line 25-27)

Introduction section

1. It would be better to include HIV morbidity and mortality data of Ethiopia. 

Response - Accepted and incorporated ( page 3,line 32-34)

2. Page 3, line 43 & Page 4, line 47: Reference 10,11 & 13, majority of patients received ART initiation within a day of diagnosis confirmed. May need to revise the sentences to show still need to reach the global target because in this study delay is defined for >7 days of diagnosis - 

Response- The sentence was having edition problem and it is also revised as follows to clearly show as there’s gap to reach WHO target. In Africa, a study conducted in Ekurhuleni District, South Africa, revealed that 54% of HIV-positive people were initiated into ART with the UTT strategy[12] . Similarly , related studies in South Africa and Zimbabwe reported that, only 40.1% and 65% of HIV-positive individuals were initiated into ART on the same day of confirmation of HIV/AIDS, respectively (page 4, line #55) 

Methods and Material section 

1. It would be better to explain ART diagnosis and ART initiation processes. It can be performed at the same center or is there any patient take ART test from different places enrolled into ART service center? Can delay in ART initiation happen due to service factors? 

Response - we try to explain about the ART diagnosis and ART initiation processes. “ All ART centers provide ART service starting from early diagnosis of HIV positive adults, immediately linking them to ART , monitoring their progress through continuous follow-up” page 6 , line 94-96

 Study Design and Period

1. Page 4, line 67: The author mentioned that study participants were 400 participants enrolled into ART from November 2016 to October 2020. But in page, line 78, the study population was selected from the record from November 2016 to December 2017. The study inclusion period is different.

Response—The study participants were selected from November 206 to December 2020 not December 2017. This is an edition problem and we revised it accordingly

( page 5, line 901 and page 6,line# 106)

Page5, line 79: Why study subjects were selected from two years periods 2016-2017? Is there any changes in ART health care system after 2017? Using 2016-2017, the result may not reflect to the current situations? 

Response- As noted above, the study participants were selected from November 206 to December 2020 not December 2017. This is an edition problem and we revised it accordingly ( page 6, line 106). 

Adult is >15 years? Response- In Ethiopian health care management system including HIV management protocol , adults were considered as those individuals whose age is ≥ 15 years of age .

2. Page 5, line 80: Why TB/HIV coinfection at presentation, Cryptococcus meningitis infection are excluded from the study?

Answer- Since the latest Ethiopian or World health organization HIV management guideline recommends differing ( extending) of ART initiation time for at least 2-8 weeks for patients with TB co-infection, and 4-6 weeks for patients with Cryptococcus meningitis to prevent serious drug reactions as well as immune reconstitution inflammatory condition. Therefore, we the exclude those patients not to have a biased finding ( which results overestimation of the true finding) . (page 7 ,line 133) and Reference # 4 and 5

 Sample size determination and Sampling technique

3. Page 5, line 85: It would be better to include the reference of using 25.8% proportion of outcome in unexposed group and Odds ratio of 1.87 for sample size calculation. Response -Accepted and referenced accordingly ( page 6,line 114))

4. Page 6, line 98: It would be better to use delay initiation of ART for outcome definition.- Response- Accepted and operationalized ( page 7 ,line131)

Result section 

1. The name of outcome variable “Delayed ART initiation” like title should be better to use constantly in tables and texts--- Accepted (page 7,121)

2. Line 160, page 10: In Table 1 and Table 2, only overall frequency and percentage of all variables should be better to be mentioned without showing the classification of delay and early initiation of ART. Because data is duplicated with Table 3. (OR) Otherwise, the information of Prevalence of late ART initiation should be better presented first in the result before the Sociodemographic characteristics of participants. What is median time to ART initiation?- 

Response - Accepted and revised according to the comment ,and data duplication is minimized ( page 10, line 192 , table 1 and page 11,line 202 ,,table 2)

What is median time- Response - The median time to ART initiation was 5.5 days with IQR( 10.00- 1.00 days) . . page 12,,line#209 

3. In Table 1, Had positive family member or know other HIV positive family member??- Response- as individuals who” Had HIV-positive family member” is the correct one and used consistently – ( table 1, page 10,line 192)

4. In Table 2, categorization of BMI is underweight or undernourished?? It would be better to use the same. Presence of Chronic Problems or chronic health problems?? It is notclear. Need foot note for BMI, OI etc --- 

Response- Accepted and modified accordingly:_ BMI categorization is underweight , normal weight and overweight and Chronic problems is modified into chronic diseases consistently )( Tale 2, page 11,line 202)

5. In Table 2 and table 3, Presence of chronic problem and History of chronic diseases?? Variables names are changed. It would be better to use constant variable name through the paper. ---

Response- Accepted and revised accordingly ( table 2, page 11 and table 3,page 13,line 

6. In Table 1 and table 3, Had positive family member or HIV-positive family member or know other HIV positive family member?? It would be better to use constant variable name through the paper. --- 

Response -As noted above Accepted and Revised accordingly ( table 1 and 3)

7. In Table 1 and table 3, Reside on Catchment area or Catchment area Residence?? It would be better to use constant variable name through the paper.

Response- Accepted and revised accordingly ( reside on catchment area is the correct one) ( Table 1 and Table 3)

8. Serial of variable names in Table 3 should be the same with that in Table 1 and Table 2. In table 3, marital status should be before education like in Table 1. The same for Reside on Catchment area. In table 3 bivariate association of BMI and delay was not included.---- Response- The order of marital status and educational level was made to be uniform across tables ( Table 1 and Table 3 ), Bivariable analysis result of BMI was included in the revised manuscript which was missed unintentionally on first draft. ( Table 1,2and 3) 

9. In Table 3, what is ** in COR and what is COR and AOR? “-“signs should be shown between numbers showing 95% CI. --- 

Response - Accepted and Revised Accordingly ( table 3,page 15)

10. Table 3, In variables of HIV-positive family member, History of OIs, History of chronic diseases and Substance, first row should be used for reference by changing row positions. It is difficult to interpret the data. It can lead to misinterpretation.- 

Response - Accepted and Revised Accordingly ( Table 3, page 14 and 15) 

11. Figure 2. How ROC curved is developed and how it is useful for your study should be written in data analysis portion. It should not be 2021 in Figure 2 title. Page 12 line 189, AOC or AUC??----

Response- The receiver operating characteristic curve (ROC curve), which is produced by plotting the true positive rate against the false positive rate of the model at various thresholds, was used to check the discriminative power of the model, and an area under the curve (AUC) value of greater than 70 was used to categorize the model as it can adequately classify individuals with and without the outcomes (Page 9, line 170)

12. Page 10, line 160: Prevalence of late ART initiation: It is better to use constant outcome variable name” Delayed ART initiation”. Prevalence?- 

Response- Accepted and Revised Accordingly( 

13. In Figure 1, I think I should not be 2021. Name of outcome variable should be constant with that in title. 

Response - The Name of the outcome variable made to be constant throughout the paper

If median time taken and proportion of delay in ART initiation are shown in texts, figure 1 may not need. – 

Response- We are interested in using a pie chart to explain the result of the study because graphs make the data more sound, easily interpretable, and attractive. In particular, if the category of the outcome is less than 3 or 4, pie charts will be recommended to present the data.,

14. Page 10, line 168: What is Predictors of loss to follow up? ----- 

Response- It is to say predictors of Delayed ART initiation , so it is revised accordingly ( page 13, line #215)

15. May to need to discuss about generalizability of the result. Why social support is recommended for reducing delay in this study? The result can reflect to the current situation---- 

Response- We incorporate the limitations of the study to provide a clear image of the generalizability of the study. Social support is recommended because the root cause why those high-risk individuals (those who do not have HIV-positive family members or substance abuse) end up with Delayed ART initiation will probably be a lack of strong family/social support, which motivates them to early engage in ART as well as retain them for a long time. Therefore ,we finally revised the recommendation as follows” Special emphasis shall be considered on those individuals with other comorbidities, and had not had other HIV-positive family member who will provide enough support on his/ her care”. ( page 18, line 302-307)

Reviewer # 3 

Abstract 

1. Page 2, line#7 - shouldn’t it be "public health institutions"?? Because you said multicenter cross-sectional study. Response- Accepted and revised ( page 1, line 10)

2. Page 2, line#10- Change bivariable in to “ simple binary logistic regression for each potential predictor would be a better usage.- 

Response- Accepted and Revised ( page 1,line #13)

3. Page 2, line #20- recommendation goes off evidences—Modified to be aligned to the finding. 

Answer- Accepted and t he recommendation was modified as ‘Special emphasis shall be considered for individuals with identified risk factors” in order to make recommendation short and concise. How ever, we initially recommend social support because as it is indicated on discussion part, the likely explanation for delayed initiation for those individuals who didn’t have other HIV-positive family member and who is abuse of substance is lack of social support from the family as well as the significant others . Therefore, we indirectly recommend as social support need to be strengthen. ( page 3,line 24)

Introduction section

1. page 2 ,line 29- what is 30 ?– Response – It is to say 30% ( page 3,line 36)

2. page 2, line 34 -What do you mean by this? 3 goals (or) 95 goals ?? 

Response- Those the sentence is modified to make short and precise, the three 95% 2030 targets of UNAIDS states 95% of HIV-infected individuals should know their status, 95 % of individuals who knew their result should get access to (ART), and 95 % of individuals on ART should achieve virology suppression .( Page 3,line 37-38)

3. page 2,line#33,37,43, 46, 48 and 61 grammar and punctuation issues were

Response- Thoroughly examine and edited by language experts and individuals who had good experience on writing scientific papers (page 3,line#39, page 4,,line 50,59,64,66)

4. Page 2, line 60 – at national level indicates which country—

Response – It is to indicate “Ethiopia”, so it is revised contextually (page 5,line 80)

Method Section

1. Page 5, line #79 - what do you mean by this? is it 2020 or 2017??- 

Answer- It is edition problem and revised as 2020(page 6,line 106) 

2. Page 5,line#79, state full words of abbreviations at initial and put in bracket if you use it before - 

Response- Accepted and revised(page 6,line 105) 

3. Page 5, line# 85- From which literature you took you sampling parameters - 

Response- We miss to cite it initially and we revised it now (page 6,line 114) 

4. Page 6,line# 95--- disclosure status of what? - 

Response- Accepted and modified as “HIV disclosure status of HIV positive patients “(page 7,line 125)

5. Page 6, line 103 and 105- Citation for reference of BMI operational definition

Response- reference for operational definitions were addressed(Page 7,line 137)

6. Page 6,line 105- status of what –

Response- Revised as “Disclosure is recorded as “yes” when one’s HIV positive status is informed to at least one individual ( page 8,line 140

7. Page 7, line 117- provide more details of selection of variables for the models and more on backward elimination technique which were stated later in the results section.-- 

Answer - The authors try to incorporate more information’ on how variables were selected for the final multivariable model and how backward elimination technique is applied to obtain significant predictors of the outcome variable( page 9, 166-170) 

 Result section

1. Page 8,140 – 400( 100%)– No need to write % ---

Response- Accepted and revised ( page 10,,line #187)

2. Page 8, table # 1 foot note (rewrite this sentence to make grammatically correct)--- 

Response- Accepted and modified accordingly ( Page 11,line #192, table 1)

3. Page 9, line#155- spelling error- 

Response- Accepted and revised ( page 11, line #198)

4. Page 10, line #160= You don't need 95% CI for your descriptive summary. Only state 156 (39%). --- 

Response- Accepted and it is deleted( page 12, 11line 207) 

5. Page 10, line 169- 173—better to take these points on analysis procedure part---

Response - As noted above we the authors try to incorporate these and other points on variable selection portion. On the other hand, we the authors consider that ideas on line# 171-173 were highly related with the result part because they entirely talks the names and categories of the variable which are identified as statistical predictors of delayed ART initiation. (page 9,line 162-166)

6. Page 12, Figure 2. Have you spell out FHCSH---

Respond- It is writing error and modified accordingly( page 16,figure 2) 

Discussion section

1. Page 13, Line 111 and 112- how it is related to the previous sentence 

Response- We just revise the explanation as follows to relate it with previous one and it is a hypothesis, not referenced.

“Giving special attention towards maternal and child health services by both global and national health policies may result in females getting stronger counseling, education, and monitoring by health care providers when compared to males, and this finally might influence women’s decision to early engage in ART” ( line 16, line#259 -263)

2. Page 13, line 120 – Citation needed— 

Response- Accepted and done accordingly ( page 17,line 271) 

3. Page 13, line 226- How?? Seems disconnected from your previous sentences.

Response- This sentence is used to indirectly indicate the root cause why patients with chronic problem would be delayed. As noted in preceding sentence, the possible explanation of delayed ART initiation for patients with other chronic diseases might be due to unjustified fear by the patients as well as by health care providers about drug-drug interactions, high pill burdens and toxicities happening as a result of treating both chronic diseases simultaneously. This suggests that health care providers professionals' knowledge of the management of chronic diseases comorbidities of HIV/AIDS may be insufficient, and patients may not receive adequate education and counseling about it. This further implies that continuous professional development of healthcare providers through expanding on job training, short term and long-term capacity building sessions needs to be strengthened. (page 17,line 275-282)

4. Page 13, line # 230,233, 236- Grammar issues 

Response- Accepted and Corrected accordingly (page 18, line 281,285,286)

5. Page 13,line # 241-seems disconnected from your previous sentence.

Response- - Accepted and we decided to delete the sentence because it states about the future consequences of delayed initiation on substance abuse patients ( page 18,line 292) 

 Recommendation and conclusion section

1. Include limitation of the study before generalization- 

Response- Accepted and we try to incorporate the main limitation of the study which has to be undertaken when we interpret and generalize the finding of the study. ( page 19, line #302)

N.B:- The English language issues were thoroughly assessed and edited by language experts working as a lecturer in university and individuals who have good experience in writing scientific papers and manuscripts. This goes to both editors and reviewers

---

## [Decision Letter · Decision Letter 1]

30 May 2022

PONE-D-21-39787R1Delayed ART initiation in “Test and Treat era” and its associated factors among adults receiving ART at Public health institution in Northwest Ethiopia:-A multicenter cross-sectional studyPLOS ONE

Dear Dr. Tesema,

Thank you for submitting your manuscript to PLOS ONE. After careful consideration, we feel that it has merit but does not fully meet PLOS ONE’s publication criteria as it currently stands. Therefore, we invite you to submit a revised version of the manuscript that addresses the points raised during the review process.

Specifically, the reviewer suggests modifying your title and further improving the English language. In addition, please provide more discussion on the ROC curve analysis. Please note that PLOS ONE does not provide copyediting or proofs of accepted manuscripts. We therefore recommend that you carefully review your manuscript and correct any English errors at this time.

We look forward to receiving your revised manuscript.

Kind regards,

Jianhong Zhou

Staff Editor

PLOS ONE

Journal Requirements:

Reviewers' comments:

Reviewer's Responses to Questions

**Comments to the Author**

1. If the authors have adequately addressed your comments raised in a previous round of review and you feel that this manuscript is now acceptable for publication, you may indicate that here to bypass the “Comments to the Author” section, enter your conflict of interest statement in the “Confidential to Editor” section, and submit your "Accept" recommendation.

Reviewer #1: All comments have been addressed

Reviewer #2: All comments have been addressed

2. Is the manuscript technically sound, and do the data support the conclusions?

Reviewer #1: Yes

Reviewer #2: Yes

3. Has the statistical analysis been performed appropriately and rigorously? 

Reviewer #1: Yes

Reviewer #2: Yes

4. Have the authors made all data underlying the findings in their manuscript fully available?

Reviewer #1: No

Reviewer #2: Yes

5. Is the manuscript presented in an intelligible fashion and written in standard English?

Reviewer #1: Yes

Reviewer #2: No

6. Review Comments to the Author

Reviewer #1: Thank you for addressing the comments and now it is a better version. Tables and figures are more nicer than the previous version.

Reviewer #2: Manuscript Number PONE-D-21-39787R1

Delayed ART initiation in “Test and Treat era” and its associated factors among adults receiving ART at Public health institution in Northwest Ethiopia:-A multicenter cross-sectional study

Thanks authors,

I agree with your response on previous my comments. Generally, the authors need to modify figure and text abbreviation, and English checking.

I only have some minor comments.

In Title, should it be public health institutions? Because you included multicenter sites of public health institutions.

I think Title should not have abbreviation such as ART.

In abstract, it would be better to have long explanation for ART, HIV, AOR,)

In abstract, some time used two decimal for AOR but some used ARO. It would be better to be the same.

In background, page 3 line 42: for Ministry of Health in Ethiopia (FMOH), is it EMOH?

In Figure 1. Pie chart. It would be better to include (n=??)

It is not clear how it is useful for ROC curve analysis. There is no any discussion and it does not mentioned it’s usefulness in the methods session.

In discussion, male were more likely to delay than female. I think the first two reasons was not clear. We could not have any evidence to support the assumption of physician’s counseling to women influencing on women decision. It would be better to include reference. And then, we could not have any information of number of pregnancy in this study. Therefore, this explanation may be irrelevant to provide the reasons behind delays of male ART seeking.

7. PLOS authors have the option to publish the peer review history of their article (what does this mean?). If published, this will include your full peer review and any attached files.

Reviewer #1: No

Reviewer #2: **Yes: **Kyaw Ko Ko Htet

---

## [Author Response · Author response to Decision Letter 1]

1 Jun 2022

. Reviewer 1

Response- We, the authors, appreciate the time and effort you dedicated to providing feedback on our manuscript and are grateful for your constructive comments, which add significant value to the betterment of our paper

 The data should be provided as part of the manuscript or its supporting information, or deposited in a public repository. For example, in addition to summary statistics, the data points behind means, medians, and variance measures should be available

 Response— Thank you for your genuine comment. Regarding the availability of the data, we try to access all the relevant data on the manuscript and its supporting file( SPSS file 1) without any restriction. Additionally, summary statics measures like mean, and median were selected based on the nature of the data( categorical vs continuous), and considering the distribution ( skewed and normally distributed) for continuous variables. So that we are trying to adhere to this assumption.

Reviewer # 2.

Initially, we, the authors, appreciate the time and effort you dedicated to providing feedback on our manuscript and are grateful for your constructive comments, which add significant value to the betterment of our paper. We have incorporated the comments and suggestions given by you. The quality of the manuscript has significantly changed after we made a strong revision based on your recommendations. Those changes are highlighted in the manuscript and below are responses that address each question and concerns

I only have some minor comments.

In Title, should it be public health institutions? Because you included multicenter sites of public health institutions.

 Response—Thank you for your marvelous comments, It is revised accordingly ( Title)

I think Title should not have an abbreviation such as ART.

 Response—Accepted and revised accordingly 

In abstract, it would be better to have a long explanation for ART, HIV, AOR,)

 Response—Accepted and revised accordingly (page 2, line #2, page 2, line #16), but for HIV, we consider it as a well-known abbreviation so writing the full term of a well-known abbreviation will not have significant value. 

In abstract, sometimes used two decimal for AOR but some used ARO. It would be better to be the same.

 Response—Thank you for the insightful comment, we revised it to make it consistent across the manuscript( page 2, lines # 16-19)

In the background, page 3 line 42: for Ministry of Health in Ethiopia (FMOH), is it EMOH?

 Response- Thank you and it is revised accordingly ( page 3, line #42)

In Figure 1. Piechart. It would be better to include (n=??)

 Response- Accepted and revised accordingly ( page 12, Figure 1)

It is not clear how it is useful for ROC curve analysis. There is no discussion and it does not mention its usefulness in the methods session.

 Response—Thank you for your genuine comment and we revised the manuscript to make it more clear. As stated in the result section, the ROC curve is used to differentiate the discrimination power of the model i.e if the graph of the ROC curve is above the diagonal line (the line where the area under the curve (AUC) value=0.5/50%), the model is enough to discriminate those who will have delayed ART initiation( True positives) with individuals who will have not delayed ART initiation ( false positives ). But if the AUR value is less than or equal to 50%, the model will not good to differentiate between Delayed ART initiation from those who have rapid ART initiation. ( page 9, lines 169-173)

In discussion, males were more likely to delay than females. I think the first two reasons were not clear. We could not have any evidence to support the assumption of physicians’ counseling women influences women decisions. It would be better to include a reference. And then, we could not have any information on several pregnancies in this study. Therefore, this explanation may be irrelevant to providing the reasons behind delays in male ART seeking. 

 Response— Thank you for your genuine comment and we revised it accordingly. besides, the followings are our views regarding the first justification of why males will have delayed ART initiation compared to females. 

The first reason is linked with the women’s decision-making power not only in their home but rather in health care services is highly influenced by anybody else. Therefore, If females were tested positive, and immediately got adequate counsel to begin ART soon, they will have a higher probability to engage in ART soon compared to males( page 16, line#251).

The second justification was deleted 

N.B:- The English language issues were thoroughly assessed and edited by language experts working as a lecturer in universities and individuals who have good experience in writing scientific papers and manuscripts. This goes to both editors and reviewers.

---

## [Decision Letter · Decision Letter 2]

24 Jun 2022

Delayed ART initiation in “Test and Treat era” and its associated factors among adults receiving antiretroviral therapy at Public health institutions in Northwest Ethiopia: A multicenter cross-sectional study

PONE-D-21-39787R2

Dear Dr. Tesema,

We’re pleased to inform you that your manuscript has been judged scientifically suitable for publication and will be formally accepted for publication once it meets all outstanding technical requirements.

Kind regards,

Jianhong Zhou

Staff Editor

PLOS ONE

Additional Editor Comments (optional):

Reviewers' comments:

Reviewer's Responses to Questions

**Comments to the Author**

1. If the authors have adequately addressed your comments raised in a previous round of review and you feel that this manuscript is now acceptable for publication, you may indicate that here to bypass the “Comments to the Author” section, enter your conflict of interest statement in the “Confidential to Editor” section, and submit your "Accept" recommendation.

Reviewer #2: All comments have been addressed

2. Is the manuscript technically sound, and do the data support the conclusions?

Reviewer #2: Yes

3. Has the statistical analysis been performed appropriately and rigorously? 

Reviewer #2: Yes

4. Have the authors made all data underlying the findings in their manuscript fully available?

Reviewer #2: Yes

5. Is the manuscript presented in an intelligible fashion and written in standard English?

Reviewer #2: Yes

6. Review Comments to the Author

Reviewer #2: Manuscript Number: PONE-D-21-39787R2

Title: Delayed ART initiation in “Test and Treat era” and its associated factors among adults

receiving antiretroviral therapy at Public health institutions in Northwest Ethiopia: A

multicenter cross-sectional study

Thanks authors. I agree with your response. I have no comments on it.

7. PLOS authors have the option to publish the peer review history of their article (what does this mean?). If published, this will include your full peer review and any attached files.

Reviewer #2: **Yes: **Kyaw Ko Ko Htet

---

## [Editor Report · Acceptance letter]

14 Jul 2022

PONE-D-21-39787R2 

Delayed ART initiation in “Test and Treat era” and its associated factors among adults receiving antiretroviral therapy at Public health institutions in Northwest Ethiopia: A multicenter cross-sectional study 

Dear Dr. Bantie:

I'm pleased to inform you that your manuscript has been deemed suitable for publication in PLOS ONE. Congratulations! Your manuscript is now with our production department. 

Kind regards, 

on behalf of

Jianhong Zhou 

Staff Editor

PLOS ONE